# Aromatherapy improves cognitive dysfunction in senescence-accelerated mouse prone 8 by reducing the level of amyloid beta and tau phosphorylation

**Michiaki Okuda**[1,2¤a]*, **Yuki Fujita**[1,2¤a], **Yuki Takada-Takatori**[3], **Hachiro Sugimoto**[1¤a¤b], **Katsuya Urakami**[4]

**1** Graduate School of Brain Science, Doshisha University, Kizugawa, Kyoto, Japan, **2** Department of Pharmacology, Graduate School of Pharmaceutical Sciences, Kyoto University, Kyoto, Kyoto, Japan, **3** Faculty of Pharmaceutical Sciences, Doshisha Women's College, Kyotanabe, Kyoto, Japan, **4** Department of Biological Regulation, School of Health Science, Tottori University Faculty of Medicine, Yonago, Tottori, Japan

¤a Current address: Green Tech Co., Ltd., Kyoto, Kyoto, Japan
¤b Current address: Faculty of Life and Medical Sciences, Doshisha University, Kyotanabe, Kyoto, Japan
* okuda.michiaki.33r@st.kyoto-u.ac.jp

**Data Availability Statement:** All relevant data are within the paper and its Supporting Information files.

## Abstract

Alzheimer's disease (AD) is a progressive neurodegenerative disease and is known to be the most common cause of dementia. We previously described the benefits of aromatherapy on the cognitive function of patients with AD utilizing various aromatic essential oils; however, its mechanism of action remains poorly understood. Consequently, in the present study, this mechanism was thoroughly evaluated employing a dementia mice model, specifically the senescence-accelerated mouse prone 8. The mice were exposed to a mixture of lemon and rosemary oil at nighttime as well as to a mixture of lavender and orange oil in the daytime for 2 months. The cognitive function of the mice was assessed before and after treatment with the aromatic essential oils using the Y-maze test. Moreover, the brain levels of amyloid beta (Aβ), abnormally phosphorylated tau, and brain-derived neurotrophic factor (BDNF) were measured following treatment. The benefits of aromatherapy on the cognitive function in mice were confirmed. It was also established that the brain levels of Aβ and abnormally phosphorylated tau were considerably lower in the aromatherapy group, while the levels of BDNF were marginally higher. These results suggest that aromatherapy employing these aromatic essential oils is beneficial for the prevention and treatment of AD.

## Introduction

Alzheimer's disease (AD) is a progressive neurodegenerative disorder and is considered to be the most common cause of dementia. The number of patients with AD continues to increase globally [1]. Although a number of AD medicines have been approved, the available agents can only temporarily delay symptom progression, and currently, no drugs can actually cure

**Funding:** The author(s) received no specific funding for this work.

**Competing interests:** The authors have declared that no competing interests exist.

the disease. More than 100 clinical trials concerning AD have been carried out in the past 15 years [2]. Furthermore, approximately 100 new drugs are presently undergoing clinical trials [3]; however, no new medicines able to actually cure AD have been reported. Thus, drug-free, complementary approaches for the prevention and treatment of dementia are essential.

Complementary health approaches, including Ayurveda [4], Unani [5], balneotherapy [6], manipulative therapy [7], music therapy [8], and animal-assisted therapy [9], are frequently utilized in combination with pharmaceuticals. Aromatherapy is also a complementary approach, which is employed in numerous fields. For instance, lavender oil has been reported to improve sleep disorders [10] and reduce anxiety [11]. On the other hand, lemon oil affects the antioxidant activity of vitamin E and improves the state of blood vessels [12]. It is noteworthy that aromatherapy has also been employed for the treatment of dementia [13–15]. We have previously reported the benefits of aromatherapy for AD [16]. A mixture of lemon and rosemary aromatic essential oils, known to activate the sympathetic nervous system, improving concentration and memory, was used in the morning, and a mixture of lavender and orange oils, which activate the parasympathetic nervous system, calming the patients' nerves, was administered in the evening. The developed therapy exhibited a significant improvement in the cognitive function of patients with dementia. Nevertheless, the mechanism of action of this kind of therapy remains poorly understood.

In the present study, we evaluated beneficial effects of the aromatherapy on the brain and cognitive function utilizing a dementia mice model the senescence-accelerated mouse prone 8 (SAMP8). SAMP8 is a substrain of senescence-accelerated mouse, which is a murine model of accelerated aging [17], displaying age-related deficits in memory and learning [18, 19]. Notably, in the AD brain, Aβ forms senile plaques (SP) [20], while phosphorylated tau forms neurofibrillary tangles (NFT) [21]. SP and NFT are widely recognized as the two major hallmarks of AD. SAMP8 mice are also known to exhibit Aβ and tau pathology in the brain [22, 23]. Hence, we administered aromatic essential oils to the mice for 2 months and assessed their cognitive function using the Y-maze test. Additionally, the motor function was investigated employing a rotarod test and a grip strength test. Subsequently, we measured the amount of Aβ, phosphorylated tau, and brain-derived neurotrophic factor (BDNF) in the mice brain following treatment. BDNF is a neurotrophic factor, which promotes growth, survival, and differentiation of neurons in the brain [24]. It is understood that the amount of BDNF is decreased in individuals with preclinical AD compared with cognitively healthy individuals [25]. In the current study, the conducted aromatherapy treatment prevented cognitive and motor function decline. It was established that brain levels of both Aβ and phosphorylated tau were considerably lower in the aromatherapy group, while the BDNF levels were somewhat higher. The obtained outcomes imply that aromatherapy can be considered as a valuable tool for the prevention or treatment of dementia, including AD.

## Materials and methods

### Reagents

The set of aromatic essential oils (lot no. 5E) used in this study were purchased from Brainmate Co. Ltd. (Tokyo Japan). The lemon essential oil was obtained by squeezing the peels of lemon (*Citrus Limon*). The rosemary essential oil was obtained by steam distillation from the leaves of rosemary (*Rosmarinus officinalis L*, ct1 Camphor). The orange essential oil was obtained by squeezing the peels of sweet orange (*Citrus sinensis*). The lavender essential oil was obtained by steam distillation from the flowers and ears of true lavender (*Lavandula angustifolia*, *L.vera*, *L. officinalis*).

## Animals

Male SAMP8/TaSlc mice (3 months of age) were obtained from Japan SLC Inc. (Hamamatsu, Japan). The mice were kept in a regulated environment (temperature 24˚C ± 3˚C; humidity 50% ± 10%; 12 h inverted light-dark cycle) and allowed uninterrupted access to food and tap water. This study was approved by the Kyoto University Animal Experimentation Committee (authorization number: 16-56-2) and the experimental procedures concerning mice and their care were carried out in accordance with the ethical guidelines of the committee. All efforts were made to minimize suffering.

## Aromatherapy treatment with essential oils

The mice were kept in individual cages in a breeding room (2.3 m × 4.5 m × 2.5 m). For administration of the aromatic essential oils, the mice were moved in their cages to the treatment rooms (2.3 m × 2.2 m × 2.5 m). In the first study, the mice were exposed to a mixture of 15 μL of lemon and 30 μL of rosemary essential oils at nighttime (from 8:00 pm to 8:00 am; group A, n = 6) or to a mixture of 30 μL of lavender and 15 μL of orange essential oils at daytime (from 8:00 am to 8:00 pm; group B, n = 6). Each oil mixture was impregnated onto a 10 cm filter paper and placed at about 30 cm from the cages. The cages ware placed evenly around the filter paper. The oils were utilized in a day-night reversal relative to humans [16], because mice are nocturnal. The mice in the control group (n = 6) were moved to the treatment room with group A, and then moved back to the breeding room without any essential oil treatment. In the second study, the mice were exposed to a mixture of 15 μL of lemon and 30 μL of rosemary essential oils at nighttime (from 8:00 pm to 8:00 am) and to a mixture of 30 μL of lavender and 15 μL of orange essential oils at daytime (from 8:00 am to 8:00 pm; group C, n = 8). Each oil mixture was impregnated onto a 10 cm filter paper and placed at about 30 cm from the cages, as in the first study. Mice in the control group (n = 9) were moved to the treatment room with group C, and then moved back to the breeding room without any treatment. The described treatments were conducted 4 days a week (on 2 consecutive days with a rest of 1–2 days) over a period of 2 months (9 weeks). This treatment period was determined by referring to the results of the studies we have conducted [26, 27]. The treatment of each group is summarized in Table 1.

## Y-maze test

The spatial working memory of the mice in all groups was assessed employing the Y-maze test at the start as well as in the 4th and the 8th week of the treatment. The Y maze has three arms, each 30 cm in length, with equal angles between them all. The maze also contains a wall, which is 12 cm high. Each mouse was placed on one arm of the maze and allowed to move freely for 8 min. Both the sequence and the number of arm entries were recorded. Spontaneous

**Table 1. Summary of the aromatherapy in the present study.**

|  |  |  | Aroma essential oil treatment | |
|  |  |  | Nighttime | Daytime |
|  | Group | n | (Lemon and Rosemary) | (Lavender and Orange) |
|---|---|---|---|---|
| The 1st study | Control | 6 | - | - |
|  | A | 6 | + | - |
|  | B | 6 | - | + |
| The 2nd study | Control | 9 | - | - |
|  | C | 8 | + | + |

alternation behavior, which is frequently used as a measure of spatial memory, was defined as sequential entry into all three arms. The percentage of spontaneous alternations was calculated according to the following equation:

Number of spontaneous alternation/(number of total arm entries − 1) × 100

## Motor function test

The ability of coordinated movements of the mice was examined utilizing a rotarod treadmill MK-670 (Muromachi Kikai, Tokyo, Japan) in the 9th week of treatment. The mice were placed on a rotating rod, which was gradually accelerated to 40 rounds per minute over 300 s. The latency to fall from the rod was recorded for a maximum of 300 s. Three trials were performed. The first trial was treated as training; therefore, it was not included in the evaluation, i.e., the mean of the second and third trials was determined. Following the rotarod test, the grip strength of the mice was also assessed using a mice grip strength meter MK-380M (Muromachi Kikai). Three trials were carried out and the mean was calculated using the obtained data.

## Protein extraction from the brain tissues

At the end of the treatment, all mice were sacrificed by cervical dislocation, and the olfactory bulb and hippocampus in the brain were removed for biochemical examination. The brain tissue was homogenized in ten volumes (w/v) of 2 × RIPA buffer (Nacalai Tesque, Inc. Kyoto, Japan) with 1 × PhosSTOP (Roche, Basel, Switzerland), and 1% protease inhibitor cocktail (Nacalai Tesque). To establish the levels of Aβ, tau, and BDNF, the homogenate was centrifuged at $15{,}000 \times g$ at 4˚C for 60 min, and the supernatant was collected as a protein extract.

## Enzyme-linked immunosorbent assay

The levels of $A\beta_{42}$ and $A\beta_{40}$ in the brain were measured using the Aβ enzyme-linked immunosorbent assay for amyloid-β (ELISA) kits (Wako Pure Chemical Industries, Inc., Osaka, Japan) and a Model 680 microplate reader (Bio-Rad Laboratories, Inc., Hercules, CA, USA) according to the manufacturer's protocol. The protein extract was diluted 10-fold with the dilution buffer included in the kit.

## Western blotting

The protein extract was mixed with an equal volume of the Tris-SDS β ME sample buffer (Cosmo Bio Co., Ltd., Tokyo, Japan) and boiled at 100˚C for 10 min. For the measurement of tau levels, the samples were electrophoresed at 200 V for 1 h on a 5%–20% polyacrylamide gel (Wako Pure Chemical). On the other hand, for the BDNF measurement, the samples were electrophoresed at 100 V for 2 h on a 15%–20% tricine gel (Wako Pure Chemical). The electrophoresed samples were transferred onto 0.45 μm polyvinylidene difluoride membranes (Merck Millipore, Billerica, MA, USA), which were run at 8 V for 60 min. Following blocking with 2.5% skim milk (Nacalai Tesque) in Tris-buffered saline containing 0.05% Tween 20 (TBS-T; Sigma-Aldrich, St. louis, MO, USA) for 1 h, the blots were incubated with anti-phosphorylated (Ser202, Thr205) tau AT8 (1:1,000 dilution; Thermo Fisher Scientific, Waltham, MA, USA), anti-tau antibody TAU-5 (1:2,000 dilution; Thermo Fisher Scientific), anti-BDNF antibody N-20 (1:200 dilution; Santa Cruz Biotechnology, Dallas, TX, USA), or anti-glyceraldehyde-3-phosphate dehydrogenase (GAPDH) antibody 14C10 (1:5000 dilution; Cell Signaling Technology, Inc., Danvers, MA, USA) overnight at 4˚C. The blots were washed three times with TBS-T for 10 min and incubated with horseradish peroxidase (HRP)-conjugated anti-mouse or anti-rabbit immunoglobulin G (1:3,000 dilution; GE Healthcare, Little Chalfont,

Buckinghamshire, UK) for 1 h at ambient temperature. Following washing with TBS-T for 10 min three times, the proteins were detected utilizing a chemiluminescent HRP substrate (Merck Millipore) and analyzed employing an image analyzer LAS4000 (GE Healthcare). The intensity of the protein band was normalized against GAPDH.

### Data analysis

The data were expressed as mean ± standard error of the mean (SEM). The three-group comparison data (Fig 1A and 1B) were evaluated using one-way ANOVA and the Dunnett test. The remaining data were analyzed by the Mann-Whitney test. The GraphPad Prism software (GraphPad Software Inc., San Diego, CA, USA) was employed for the analyses, and $P < 0.05$ was considered statistically significant.

## Results and discussion

To confirm the benefits established in our previous study [16] considering cognitive deficits, male SAMP8 mice were exposed to four essential oils for 9 weeks.

In the first experiment, the aroma essential oils for daytime and nighttime were treated independently. During the treatment, the spatial working memory of the mice was assessed using the Y-maze test (Fig 1). At the start of the treatment, the percentage of spontaneous alternations in each group was approximately 73.4% (73.5% ± 2.9% for the control group, 73.4% ± 2.5% for group A, and 73.4% ± 2.6% for group B). Moreover, after 8 weeks, the percentage of spontaneous alternations of all the groups decreased with age, and there was no notable difference between the considered groups (55.7% ± 2.6% for the control group, 57.8% ± 3.1% for group A, and 61.8% ± 1.9% for group B. $P = 0.28$, one-way ANOVA, Fig 1A). There was also no change in the total number of arm entries between the groups (Fig 1B).

In the second experiment, the aroma essential oils for both daytime and nighttime were administered together. At the start of the treatment, the percentage of spontaneous alternations in each group was approximately 71% (70.8% ± 2.1% for the control group and 70.8% ± 1.9% for group C). In the control group, the percentage of spontaneous alternations decreased with age to 52.6% ± 1.9% after 8 weeks (Fig 1C). On the other hand, in the aromatherapy group, the decline of percentage of spontaneous alternations was suppressed (64.4% ± 2.2%) and the difference between the two groups was significant ($P = 0.001$, Mann-Whitney test). There was no change in the total number of arm entries in either group ($P = 0.14$, M-W test, Fig 1D). The obtained outcomes imply that aromatherapy ameliorates cognitive dysfunction in SAMP8 mice.

A decline in the walking speed as well as in the handgrip strength is reportedly associated with an increases risk of AD [28]. Hence, in the second experiment, we investigated the walking ability and grip strength of the mice following aromatherapy in the 9th week. In the rotarod test (Fig 1E), the mice in group C exhibited a considerably increased latency to fall in comparison to the control mice (145 ± 24 s for the control group, 216 ± 17 s for group C, $P = 0.03$, Mann-Whitney test). Furthermore, in the grip strength test Fig (Fig 1F), group C displayed noticeably increased muscle power compared to the control mice ($P = 0.046$, Mann-Whitney test). The above results indicate that the mice in the aromatherapy group maintained a higher motor function than the control mice.

Following the treatment in the second study, we measured the amount of Aβ in the mice brains using ELISA (Fig 2). In this assessment, we utilized the hippocampus, which is the part of the brain known to be involved in the memory function, and the olfactory bulb, which is associated with the olfactory function. The aromatherapy group exhibited a notable reduction in the amount of both $A\beta_{42}$ and $A\beta_{40}$ in the hippocampus compared with the control group ($A\beta_{42}$; 271.8 ± 5.2 pmol/g-tissue for the control group and 218.9 ± 11.6 pmol/g-tissue for the

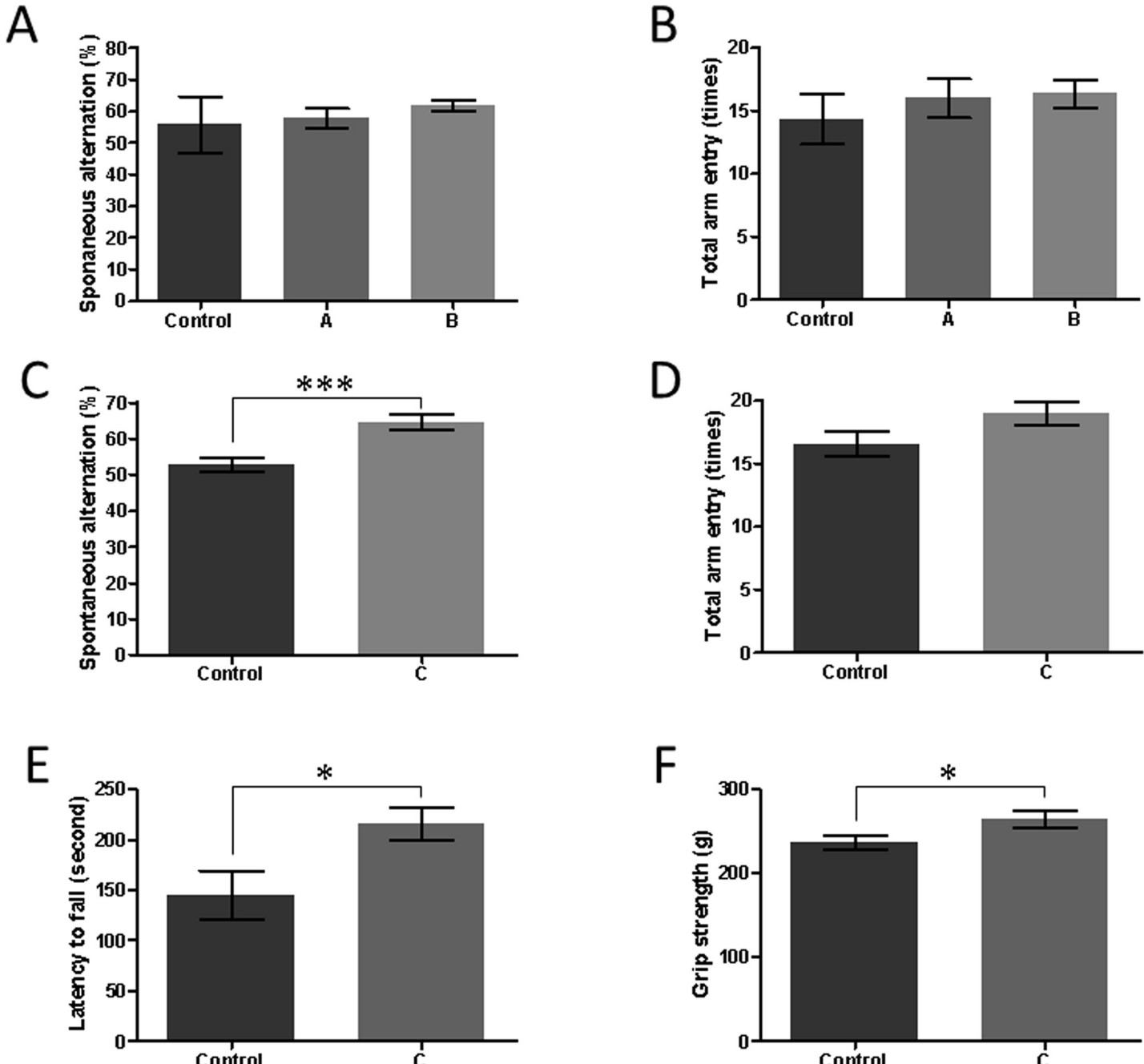

**Fig 1. Beneficial effects of aromatherapy on the cognitive dysfunction and motor function in the SAMP8 mice.** The results of the first experiment are shown in A and B, while the outcomes of the second experiment are presented in C-F. (A) Percentage of spontaneous alternation behavior in the Y-maze test. (B) The total number of arm entries in the Y-maze test. Mean ± SEM, n = 6 in each group. (C) Percentage of spontaneous alternation behavior in the Y-maze test. ***$P$ = 0.001, Mann-Whitney test. (D) The total number of arm entries in the Y-maze test. (E) Latency to fall in the rotarod test. *$P$ = 0.03, Mann-Whitney test. (F) Grip strength. *$P$ = 0.046, Mann-Whitney test. Mean ± SEM, n = 9 in the control group and n = 8 in group C.

aromatherapy group, $P$ = 0.0016, A$\beta_{40}$; 415.6 ± 35.6 pmol/g-tissue for the control group and 330.3 ± 20.2 pmol/g-tissue for the aromatherapy group $P$ = 0.046, Fig 2A and 2B). Conversely, the amount of A$\beta$ in the olfactory bulb of the aromatherapy group was marginally lower than that of the control group; however, it was not significantly different (Fig 2C and 2D).

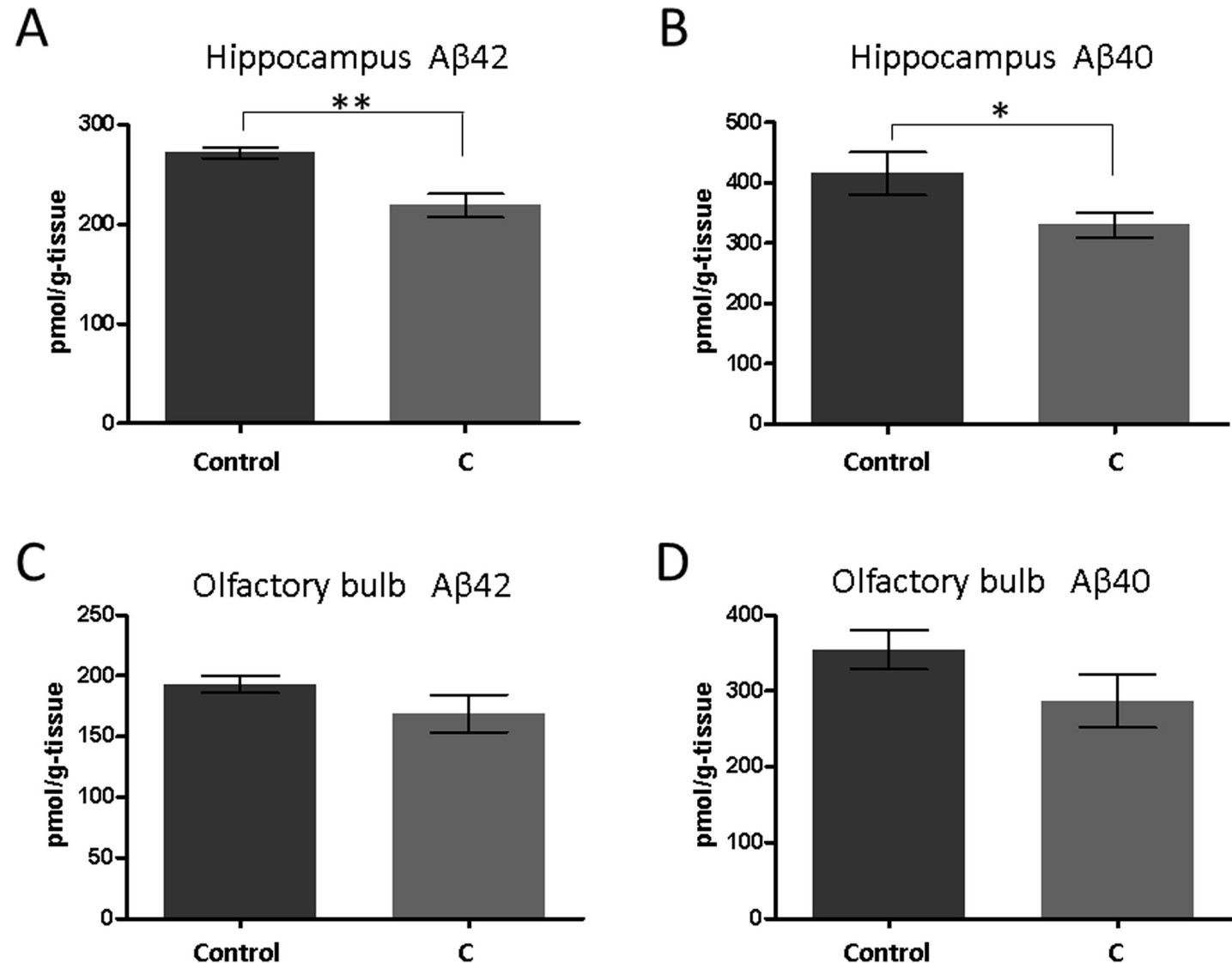

**Fig 2.** The amounts of $A\beta_{42}$ and $A\beta_{40}$ in the hippocampus (A, B) and the olfactory bulb (C, D) of the SAMP8 mice determined by ELISA. Mean ± SEM. $^{**}P = 0.0016$ and $^{*}P = 0.046$, Mann-Whitney test, n = 9 or 8.

We determined the amount of abnormally phosphorylated tau (AT8) in the brain following the aromatherapy treatment using Western blotting (Fig 3). The aromatherapy group exhibited considerably less abnormally phosphorylated tau in comparison with the control group both in the hippocampus and the olfactory bulb (Fig 3A–3C). On the other hand, there was no significant difference between the groups in the amount of total tau (TAU-5) in both tissues (Fig 3B and 3D). These outcomes indicate that aromatherapy ameliorates cognitive dysfunction in SAMP8 mice by inhibition of the Aβ production and tau phosphorylation.

The levels of BDNF in the brain (Fig 4) were also assessed. Two major forms of BDNF exist: the precursor form (pro-BDNF) and the secreted form (mature BDNF). Both of them can be effectively quantified by Western blotting. The amount of each type of BDNF in the hippocampus was analogous in each group (Fig 4A and 4B). Conversely, the amount of mature BDNF in the olfactory bulb in the aromatherapy group was marginally larger than that of the control group; however, it was not significantly different (Fig 4C and 4D).

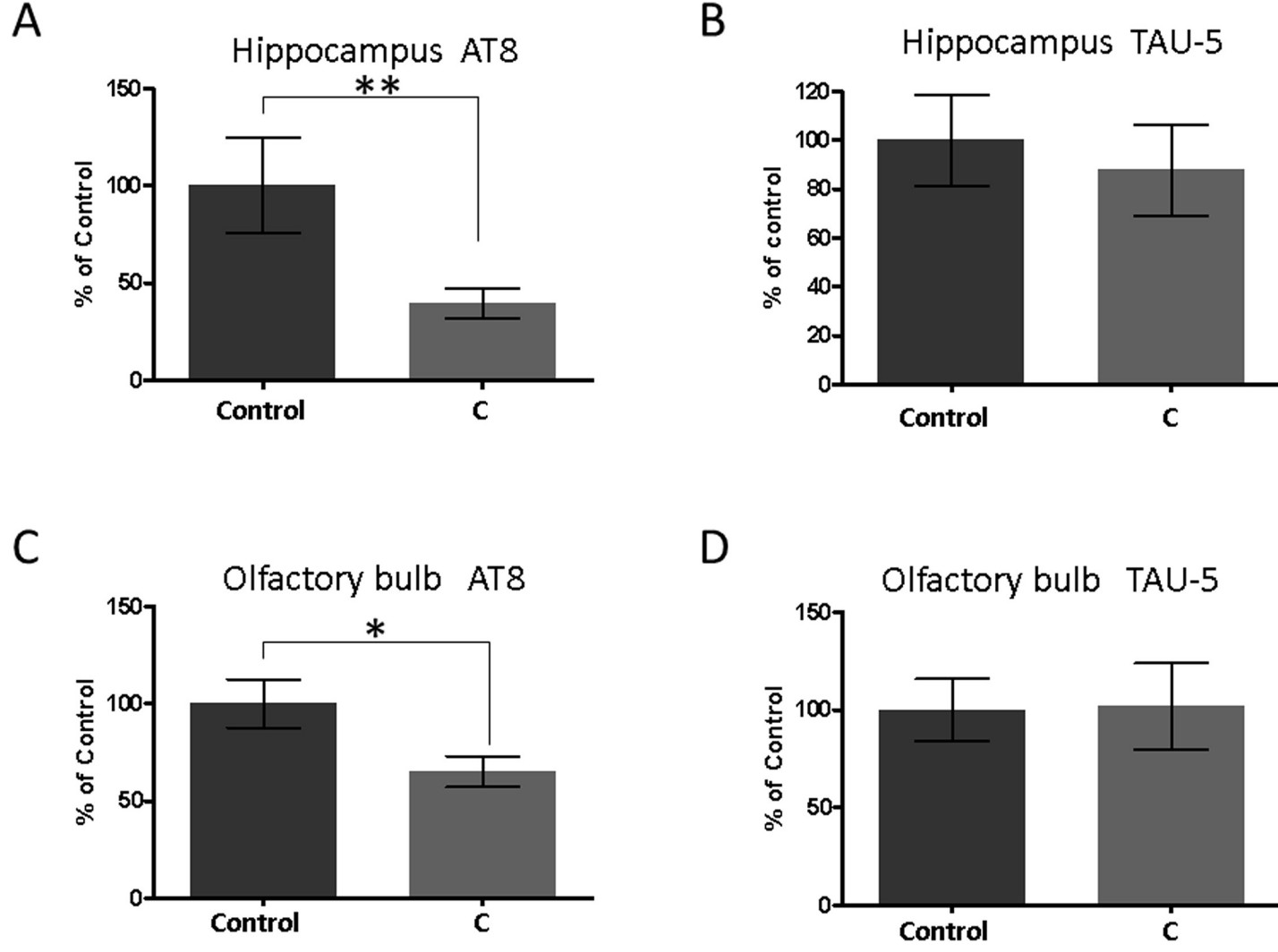

**Fig 3.** The amount of abnormally phosphorylated tau (AT8) and total tau (TAU-5) in the hippocampus (A, B) and the olfactory bulb (C, D) detected by Western blotting. Mean ± SEM, n = 9 or 8. **$P$ = 0.005, *$P$ = 0.011, Mann-Whitney test. Western blotting images of these data are shown in S1 Fig.

We have previously reported that aromatherapy enhanced cognitive function in patients with AD [16]. However, the mechanism of action remains poorly understood. Consequently, in the current study, we investigated the mechanism utilizing a dementia mice model. In the first instance, we confirmed the beneficial effects of aromatherapy on the cognitive dysfunction in SAMP8 mice (Fig 1).

In the experiments described herein, we used young (3–5 months of age) SAMP8 mice. It was established that the SAMP8 mice adequately mimicked the early state of dementia, because the spontaneous alternation rates in the Y-maze test decreased with age (from 3 to 5 months of age) [26, 27]. Hence, the obtained results imply that aromatherapy exhibits certain preventive effects on dementia. We subsequently measured the amount of Aβ, phosphorylated tau, and BDNF in the brain (Figs 2–4). These factors are all closely associated with the onset and progression of AD. Aβ and tau pathologies are key features of AD [20, 21] and can be observed over a decade prior to the onset of AD [29]. Decreased amounts of BDNF have also been noted before the onset of this disease [25]. In the present study, we determined that in the

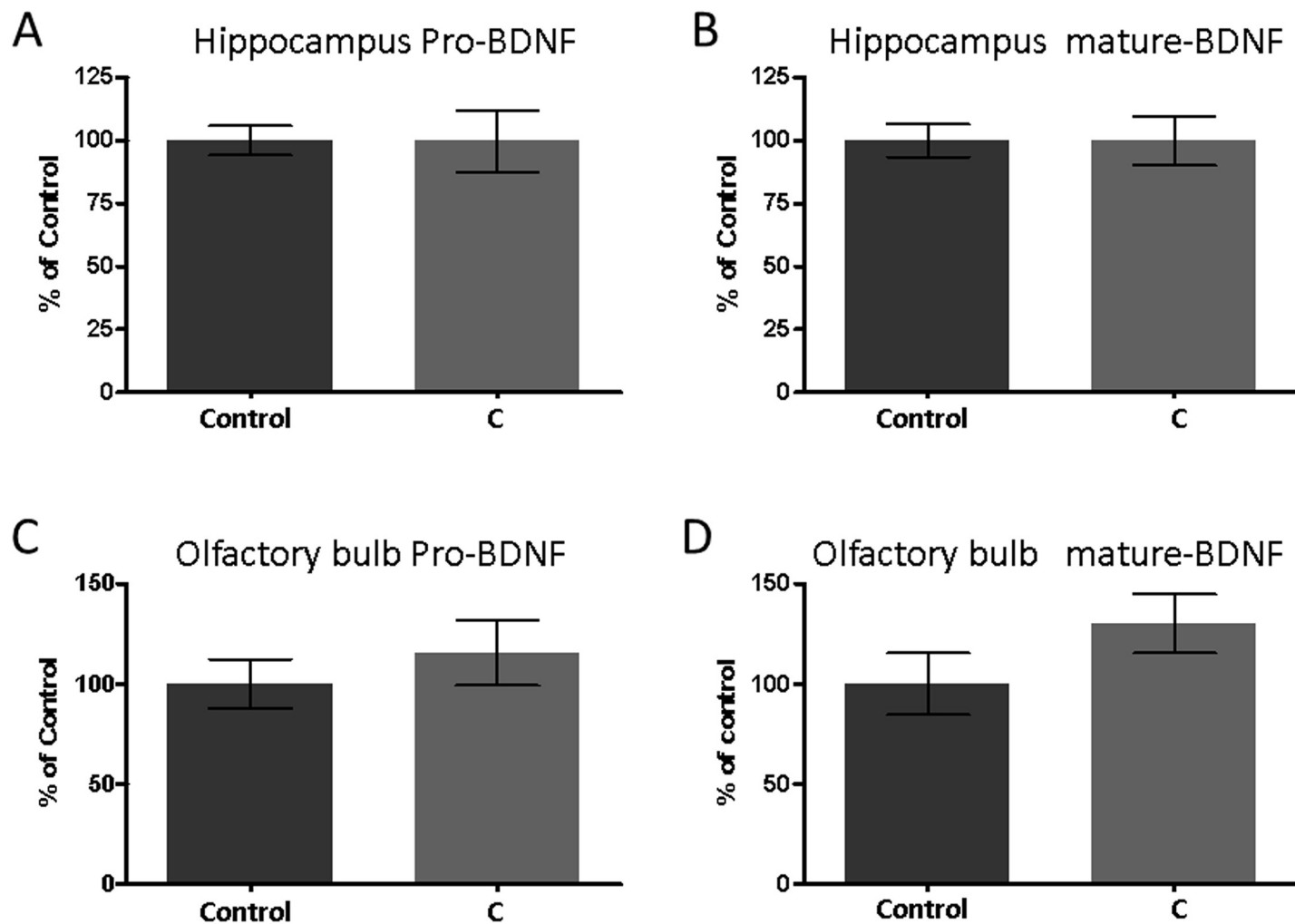

**Fig 4.** The amount of the precursor form of BDNF (pro-BDNF) and the secreted form of BDNF (mature BDNF) in the hippocampus (A, B) and the olfactory bulb (C, D) detected by Western blotting. Mean ± SEM, n = 9 or 8. Western blotting images of these data are shown in S1 Fig.

aromatherapy group, the level of Aβ was lower in the hippocampus (Fig 2A and 2B), the amount of phosphorylated tau was lower in both the hippocampus and olfactory bulb (Fig 3A–3C), while the level of BDNF was higher in the olfactory bulb (Fig 4C and 4D). Based on these outcomes, we considered various pathways, which may affect the brain levels of these factors. Firstly, some active ingredients of the aromatic essential oils may be absorbed by the brain to act directly on the neural cells. Such direct action on the neural cells could explain the elevated levels of BDNF in the olfactory bulb, which is closer to the olfactory epithelium than the hippocampus. Secondly, olfactory stimulation by aromatic essential oils may be transmitted to the brain to restore neural cells. The inhibitory effects of aromatherapy on tau phosphorylation, displayed in both brain regions, may be a consequence of this mechanism. Thirdly, the balance of the sympathetic/parasympathetic nerves may be altered upon exposure to aromatic essential oils, which may affect thermoregulation and blood flow promotion. The increase in the blood flow may rationalize the lower levels of Aβ in the hippocampus. It is known that cilostazol, which is an antiplatelet agent increasing blood flow, promotes drainage of cerebrovascular Aβ. Administration of this drug has been shown to result in cognitive improvement in patients with AD [30, 31].

The mice in the aromatherapy group maintained higher motor function than those in the control group (Fig 1E and 1F). Motor dysfunction can be associated with cognitive dysfunction in AD. It has been determined that middle-aged adults with slow-walking and a weak handgrip have a >2.5 times higher risk of developing AD [28]. Aromatherapy has been demonstrated to improve motor function. For instance, in stroke patients, acupressure combined with aromatherapy enhanced shoulder power in comparison to acupressure alone [32]. Reduced muscular power leads to decreased blood flow in the entire body, including the brain. Decreased cerebral blood flow results in the lack of oxygen and nutrition in the brain, which in turn causes deterioration of neurological functions. Thus, direct stimulation of the brain and successive improvement of the motor function by aromatherapy may exhibit a synergistic effect on the development of cognitive function.

In AD, olfactory dysfunction precedes memory and learning deficits [33]. Moreover, in early AD, NFT is initially seen in the entorhinal cortex and olfactory bulb, and subsequently spreads to the cerebral limbic systems and the cerebral cortex [34]. In the case of the APP/PS1 transgenic mice, the olfactory dysfunction precedes the memory dysfunction, and the Aβ deposition spreads in the following order: olfactory epithelium, olfactory bulb, entorhinal cortex, and hippocampus [35]. This order is analogous to the pathway of olfactory stimulation. Additionally, denervation of the olfactory epithelium by washing with detergent has been shown to lead to a decrease in the Aβ deposition in the brain [36]. These studies imply that the brain areas involved in the olfactory function are closely related to the onset of AD. Furthermore, it can be deciphered that the therapeutic approaches acting on the olfactory nervous system could be effective in the prevention of AD.

Recent reports on aromatherapy have investigated its mechanism of action in the brain. For example, lavender oil displays anxiolytic effects in mice via serotonergic neurotransmission [37]. Rosemary oil has been shown to improve learning and memory in mice via antioxidant effects [38], while orange oil induces neurite outgrowth in cultured cells [39]. In addition, the present study confirms that lemon, rosemary, lavender, and orange aromatic essential oils result in lower levels of Aβ and phosphorylated tau. Moreover, olfactory stimulation by these oils leads to higher levels of BDNF (Fig 4). The detailed molecular basis remains ambiguous, and further research is necessary. Furthermore, humans and mice exhibit different olfactory sensitivity and odor preference; thus, further research is needed for human clinical applications.

## Conclusions

We examined the benefits of aromatherapy on AD as well as its mechanism of action employing a dementia mice model, namely SAMP8. We exposed the mice to a mixture of lemon and rosemary oils at nighttime, and a mixture of lavender and orange oils during the daytime for 2 months. The benefit of aromatherapy on the cognitive function was confirmed by the assessment using the Y-maze test. We also measured the levels of Aβ, abnormally phosphorylated tau, and BDNF in the brain following aromatherapy. We found that the brain levels of Aβ and abnormally phosphorylated tau were considerably lower in the aromatherapy group, and the BDNF levels were marginally higher, implying that aromatherapy using these aromatic essential oils is beneficial for the prevention and treatment of AD.

## Supporting information

**S1 Fig. Western blotting images of Figs 3 and 4.**
(PDF)

## Author Contributions

**Conceptualization:** Katsuya Urakami.

**Data curation:** Michiaki Okuda, Yuki Fujita.

**Formal analysis:** Michiaki Okuda, Yuki Fujita.

**Methodology:** Michiaki Okuda, Hachiro Sugimoto, Katsuya Urakami.

**Supervision:** Hachiro Sugimoto, Katsuya Urakami.

**Writing – original draft:** Michiaki Okuda.

**Writing – review & editing:** Michiaki Okuda, Yuki Takada-Takatori, Hachiro Sugimoto, Katsuya Urakami.

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
