## [Decision Letter · Decision Letter 0]

20 Jul 2020

PONE-D-20-09948

Aromatherapy improves cognitive dysfunction in senescence-accelerated mouse prone 8 by reducing the level of amyloid beta and tau phosphorylation.

PLOS ONE

Dear Dr. Okuda:

Thank you for submitting your manuscript to PLOS ONE. After careful consideration, we feel that it has merit but does not fully meet PLOS ONE’s publication criteria as it currently stands. Therefore, we invite you to submit a revised version of the manuscript that addresses the points raised during the review process.

Reviewers questioned the use of only male mice in the study. Some problems with statistical analyses of the data was also noted.

We look forward to receiving your revised manuscript.

Kind regards,

Hemant K. Paudel

Academic Editor

PLOS ONE

Journal Requirements:

2. We understand that you purchased essential oils for this study. In your Methods section, please provide additional regarding the source of this material. Please provide the further details about the purchased items including lot number, source origin, description of appearance, whether the company provided any purity or yield measurements, and whether any chemical characterization was performed.

Please also provide further detail about oil exposure procedure, stating when it was added to the disks before exposure and the distance of the disks from cage.

3.Thank you for including your ethics statement:  "The experimental procedures concerning mice and their care in this study were carried out in accordance with the ethical guidelines of the Kyoto University Animal Experimentation Committee.".   

Please amend your current ethics statement to confirm that your named ethics committee specifically approved this study.

For additional information about PLOS ONE submissions requirements for ethics oversight of animal work, please refer to http://journals.plos.org/plosone/s/submission-guidelines#loc-animal-research  

4.PLOS ONE now requires that authors provide the original uncropped and unadjusted images underlying all blot or gel results reported in a submission’s figures or Supporting Information files. This policy and the journal’s other requirements for blot/gel reporting and figure preparation are described in detail at https://journals.plos.org/plosone/s/figures#loc-blot-and-gel-reporting-requirements and https://journals.plos.org/plosone/s/figures#loc-preparing-figures-from-image-files. When you submit your revised manuscript, please ensure that your figures adhere fully to these guidelines and provide the original underlying images for all blot or gel data reported in your submission. See the following link for instructions on providing the original image data: https://journals.plos.org/plosone/s/figures#loc-original-images-for-blots-and-gels.

Reviewers' comments:

Reviewer's Responses to Questions

**Comments to the Author**

1. Is the manuscript technically sound, and do the data support the conclusions?

Reviewer #1: Yes

Reviewer #2: Yes

2. Has the statistical analysis been performed appropriately and rigorously? 

Reviewer #1: Yes

Reviewer #2: Yes

3. Have the authors made all data underlying the findings in their manuscript fully available?

Reviewer #1: Yes

Reviewer #2: Yes

4. Is the manuscript presented in an intelligible fashion and written in standard English?

Reviewer #1: Yes

Reviewer #2: Yes

5. Review Comments to the Author

Reviewer #1: This is a good article. There are a few lacunae. Why was Y maze test specifically chosen. What would be the approximate amounts of aromatic oils required if the subject was human. I recommend that the article be published after minor revision

Reviewer #2: The manuscript entitled “Aromatherapy improves cognitive dysfunction in senescence-accelerated mouse prone 8 by reducing the level of amyloid beta and tau phosphorylation”, by Okuda et al. studied the effect of aromatherapy in three-month-old male senescence-accelerated mouse prone 8 model. The research is novel and interesting. The manuscript is well written. Most conclusions are supported by the results presented. Authors studied AD manifestations both at protein and behavioral levels, correlating the science to the previous research published by the authors in AD patients. There are some minor concerns which if addressed, would significantly improve clarity of the work to the readers.

Minor Comments:

1. In the present study, authors claim to evaluate the “mechanism of action” of aromatherapy, but no mechanism of action is proposed or discussed at a molecular level. Please clarify?

2. Please provide reasoning for only selecting male mice in this study.

3. How was the duration of treatment (9 weeks) selected?

4. Lines 212-214 state: “in the aromatherapy group, the percentage of spontaneous alternations remained at the same level as at the start of treatment (64.4% ± 2.2%). The difference between the two groups was significant (P = 0.001, Mann-Whitney test). There was no change in the total number of arm entries in either group (P = 0.14, M-W test, Fig. 1D).”

These sentences are unclear. Also, 64.4% ± 2.2% does not seem to be at the same level with the values reported for control. Please reword these sentences for clarity.

5. Providing a table for control and treatment groups, with days and type of treatment would be beneficial to the readers.

6. Manuscript contains some typographical errors.

6. PLOS authors have the option to publish the peer review history of their article (what does this mean?). If published, this will include your full peer review and any attached files.

Reviewer #1: No

Reviewer #2: No

---

## [Author Response · Author response to Decision Letter 0]

27 Aug 2020

To editor;

1. The format of our manuscript was amended by referring to The PLOS ONE style templates.

2.The information of the essential oils and the exposure procedure was added to the Materials and Method section.

3. The ethics statement was amended.

4. The blot images for figures 3 and 4 are in supporting information Fig S1.

To Reviewer #1;

Why was Y maze test specifically chosen? 

In our previous study we have tried the Morris water maze in the same type of mice (Reference 26), but it was difficult to compare between groups because of the considerable variation in the data between each mouse. In addition, we have also tried the novel object recognition test in SAMP8 mice, but we could obtain no evaluable data because the mice were very rough and restless. Therefore, in the present study, we specifically chose the Y-maze test.

What would be the approximate amounts of aromatic oils required if the subject was human?

We think it very difficult to estimate the required amounts of oils in human at the moment because olfactory sensitivity varies by species.

To Reviewer #2;

1. In the present study, authors claim to evaluate the “mechanism of action” of aromatherapy, but no mechanism of action is proposed or discussed at a molecular level. Please clarify?

In our previous study only cognitive function was tested （Reference 16）. But, in the present study, we newly measured the brain levels of some proteins such as tau and Aβ, so we used the word “mechanism” in the manuscript. We modified this expression more concretely in the introduction (line 66-).

2. Please provide reasoning for only selecting male mice in this study.

We have usually used male mice in our studies because females have a sexual cycle and it may affect mental state and behavior of mice, leading to data variability in behavioral tests. And, reportedly, olfactory ability is affected by circulating sex hormones in mice (Kass et al., Sci Rep. 2017 Apr 26;7:45851. doi: 10.1038/srep45851). So, it may be more difficult to evaluate the effects of aromatherapy in female because of the changes in olfactory function due to the sexual cycle. However, it is interesting to do the comparative study between male and female in the future.

3. How was the duration of treatment (9 weeks) selected?

At least we found that a significant cognitive decline was seen in 2-3 months in young SAMP8 mice evaluated using Y-maze test from our previous studies (References 26 and 27). Therefore, we set the treatment periods to 9 weeks in the present study.

4. Lines 212-214 state: “in the aromatherapy group, the percentage of spontaneous alternations remained at the same level as at the start of treatment (64.4% ± 2.2%). The difference between the two groups was significant (P = 0.001, Mann-Whitney test). There was no change in the total number of arm entries in either group (P = 0.14, M-W test, Fig. 1D).” These sentences are unclear. Also, 64.4% ± 2.2% does not seem to be at the same level with the values reported for control. Please reword these sentences for clarity.

We modified the expression of sentences in lines 221-224 to match the data.

5. Providing a table for control and treatment groups, with days and type of treatment would be beneficial to the readers.

We added the ｔable for the summary of treatment (Table 1).

6. Manuscript contains some typographical errors.

We corrected the manuscript being checked by a native speaker.

---

## [Editor Report · Decision Letter 1]

25 Sep 2020

Aromatherapy improves cognitive dysfunction in senescence-accelerated mouse prone 8 by reducing the level of amyloid beta and tau phosphorylation.

PONE-D-20-09948R1

Dear Dr. Okuda:

We’re pleased to inform you that your manuscript has been judged scientifically suitable for publication and will be formally accepted for publication once it meets all outstanding technical requirements.

Kind regards,

Hemant K. Paudel

Academic Editor

PLOS ONE
---

## [Editor Report · Acceptance letter]

1 Oct 2020

PONE-D-20-09948R1 

Aromatherapy improves cognitive dysfunction in senescence-accelerated mouse prone 8 by reducing the level of amyloid beta and tau phosphorylation. 

Dear Dr. Okuda:

I'm pleased to inform you that your manuscript has been deemed suitable for publication in PLOS ONE. Congratulations! Your manuscript is now with our production department. 

Kind regards, 

on behalf of

Dr. Hemant K. Paudel 

Academic Editor

PLOS ONE